
# Impact of second trip echoes for space-borne high PRF nadir-looking W-band cloud radars

Alessandro Battaglia[1,2,3]

[1]Politecnico of Torino, Turin, Italy
[2]University of Leicester, Leicester, UK
[3]National Centre for Earth Observation, Leicester, UK

**Correspondence:** Alessandro Battaglia
ab474@le.ac.uk

**Abstract.** The appearance of second trip echoes generated by mirror images over the ocean and by multiple scattering tails in correspondence of deep convective cores has been investigated for space-borne nadir-looking W-band cloud radar observations. Examples extracted from the CloudSat radar are used to demonstrate the mechanisms of formation and to validate the modeling of such returns. A statistical analysis shows that, for CloudSat observations, second trip echoes are rare and appear only above

20 km (thus easy to remove). CloudSat climatology is then used to estimate the occurrence of second trip echoes in the different configurations envisaged for the operations of the EarthCARE radar, which will adopt pulse repetition frequencies much higher than the one used by the CloudSat radar in order to improve its Doppler capabilities. Our findings predict that the presence of such echoes in EarthCARE observations cannot be neglected: in particular, over the ocean, mirror images will tend to populate the EarthCARE sampling window with a maximum frequency at its upper boundary. This will create an additional

fake cloud cover in the upper troposphere (of the order of 3% at the top of the sampling window and steadily decreasing moving downwards) and, in much less frequent instances, it will cause an amplification of signals in areas where clouds are already present. Multiple scattering tails will produce also second trip echoes but with much lower frequencies: less than one profile out of 1000 in the Tropics and practically no effects at high latitudes. At the moment Level-2 algorithms of the EarthCARE radar do not account for such occurrences. We recommend to properly remove these second trip echoes and to correct for reflectivity

enhancements, where needed. More generally this work is relevant for the design of future space-borne Doppler W-band radar missions.

## 1  Introduction

Thanks to their unique capability of penetrating and profiling cloud and precipitation space-borne mm-radars are becoming an essential component of the Global Observing System. This has been widely demonstrated by the Ka-band radar on board

the GPM mission (Skofronick-Jackson et al., 2016) and by the CloudSat Cloud Profiling Radar (CPR) (Tanelli et al., 2008). New missions are at the horizon: the ESA-JAXA EarthCARE (EC) mission (Illingworth et al., 2015) will deploy a W-band Doppler systems whereas NASA is currently in Phase 0 for the ATMOS mission, which will carry dual-frequency (Doppler) systems with frequencies ranging from Ku to W-band in different orbits (Battaglia et al., 2020; Kumjian et al., 2020). In the





next decade Doppler observations will offer a new perspective in process studies [e.g. in the characterization of convective motions, Kollias et al. (2018)) and are expected to improve microphysical retrievals (e.g. in rain by providing a robust estimate of raindrop size (Mróz et al., 2020; Mason et al., 2017) or in snow by distinguishing between rimed and fluffy snowflakes (Mason et al., 2018)]. Doppler from low Earth orbiting satellites is challenging because the high platform motion combined

with the finite antenna beamwidth introduces significant Doppler broadening which implies a significant reduction in the medium coherency time. Since Doppler radar measurements are derived from coherent measurements of the phase shifts of successive radar pulses, the time between different pulses, known as the pulse repetition interval (PRI), which is the inverse of the pulse repetition frequency (PRF), must be significantly shorter than the decorrelation time of the medium. In order to fulfill this requirement radar systems with increasingly high PRFs have been considered (which is also beneficial for the

augmented Nyquist range). For instance the EC CPR is expected to operate a PRF between 6.0 and 7.2 kHz (much higher than the CloudSat PRF ranging from 3.7 to 4.4 kHz). However, these benefits come at a price: the reduced unambiguous range, $r_u$, i.e. the maximum distance at which a target can be located to ensure that the backscattered power from that target corresponds to the latest transmitted pulse. This quantity can be computed as $r_u = \dfrac{c}{2 \times PRF}$. The tradeoff in the selection of the PRF, also referred to as the "Doppler dilemma", has been known since the early days of radar meteorology (Doviak

and Zrnić, 2006), particularly in association with ground-based precipitation scanning radars. For such systems the need of monitoring precipitation at hundreds of kilometers has posed strong constraints in the PRF selection and mitigating techniques have been implemented, e.g Torres et al. (2004). From space the atmosphere with significant cloud and precipitation targets appear much thinner, with clouds very rarely reaching above 20 km, even in the Tropics. A PRF of 7.5 kHZ, which corresponds to $r_u \approx 20 \, km$, seems right at the edge to avoid significant numbers of second trip echoes. However two situations, both caused

by multiple scattering (MS) events (for a thorough review of the topic see Battaglia et al. (2011)), that can produce significant returns from ranges much longer than the surface echo and exceeding the unambiguous range must be considered.

1. Mirror images, i.e. virtual images of atmospheric targets that appear to come from below the surface (Li and Nakamura, 2002; Meneghini and Atlas, 1986) and are associated to a reflection of energy from the surface to the target and back to the radar via a second reflection at the surface; mirror images are more prominent over ocean surfaces and with incidence

close to nadir.

2. MS tails (Battaglia and Simmer, 2008; Battaglia et al., 2010, 2014, 2016), i.e. virtual images that appear to be produced below a thick and highly scattering medium and are caused by the bouncing of the electromagnetic radiation within the scattering layer itself. For cm and mm-radars this typically occurs in correspondence to deep convection and in presence of large and dense ice particles. A remarkable example of such phenomenon was observed in CloudSat (Battaglia et al.

(2010), Fig. 21) with a MS tail above -15 dBZ visible all the way from the top of the deep convection located at 18.5 km to 25 km (the maximum sampling height of the CloudSat CPR).

In this work we explore the impact of these two phenomena by first investigating their frequency in the CloudSat database and then by exploiting the CloudSat dataset to simulate their occurrence in the upcoming EC mission. Sect. 2 provides a description of the mirror images and the MS tails and their modelling, a discussion of the mechanism how these signals produce and are



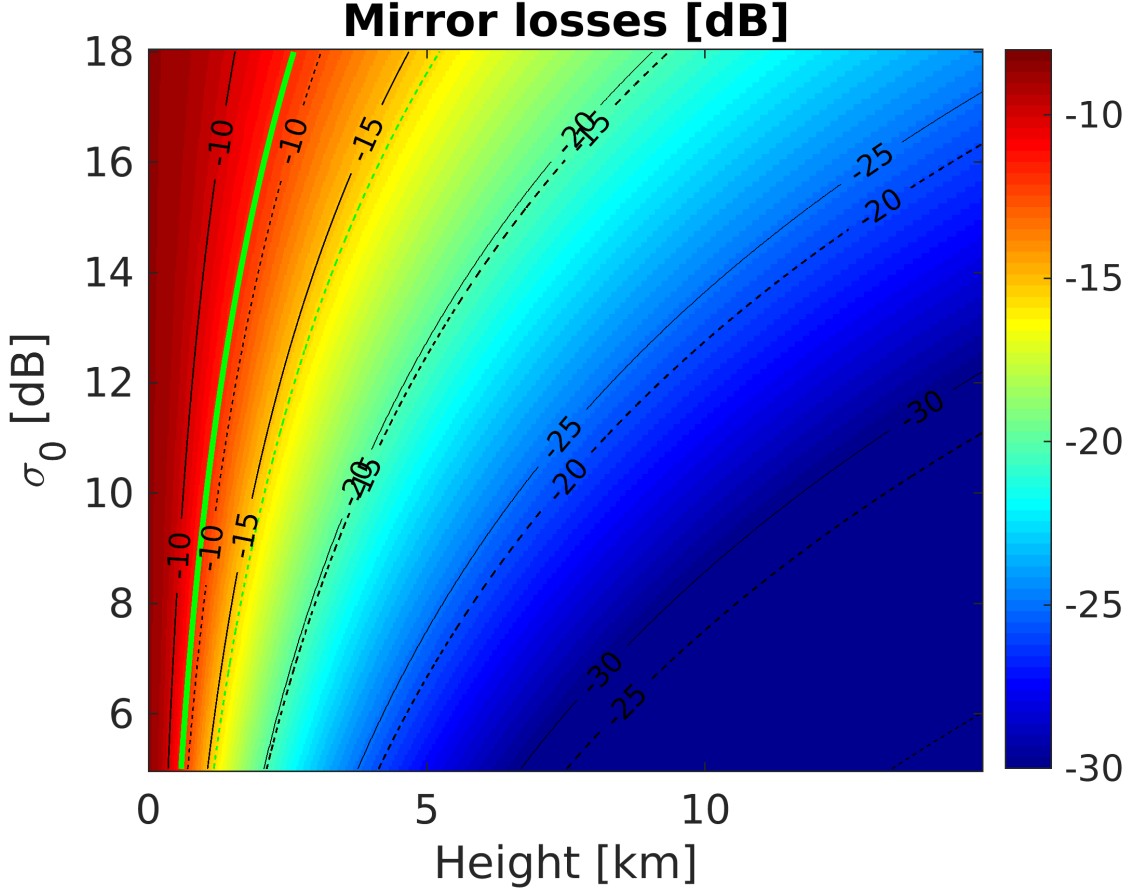

**Figure 1.** Mirror loss factor for the EarthCARE (color shadings and black continuous contour lines) and the CloudSat (black dashed contour lines) 94 GHz Cloud Profiling radars for a water surface at $10°C$. The continuous (dashed) green line corresponds to the values where the two addends in the denominator of the third term on the right hand side of Eq. (1) are equal for the EarthCARE (CloudSat) configuration.

expected to produce second trip echoes for the Cloudsat and EC CPR, plus examples extracted from the CloudSat database. In Sect. 4 a full year of CloudSat data is exploited to predict the occurrence of second trip echoes associated to mirror mages and to multiple scattering tails for the EC CPR. Conclusions and future work are presented in Sect. 5.

## 2  Mirror images and their simulation

5    The modelling of the return of mirror images is thoroughly described in Meneghini and Atlas (1986); Meneghini and Kozu (1990). The return power from the mirror, $P_r(r_m)$ appears to come from a range $r_m = r_t + 2H_t$ where $r_t$ is the range of the





target and $H_t$ its height above the surface and is related to the return power from the target itself, $P_r(H_t)$, by:

$$10\log_{10}\left[P_r(r_m)\right] = 10\log_{10}\left[P_r(r_t)\right] - 4\,A_{surface\rightarrow target} + 10\log_{10}\left[\frac{(H_{sat}-H_t)^2\Gamma^4\sigma_0}{\sigma_0 H_{sat}^2 + 11.04\Gamma^2\frac{H_t^2}{\theta_{3dB}^2}}\right] \tag{1}$$

where $H_{sat}$ is the height of the satellite, $\theta_{3dB}$ the antenna 3 dB beamwidth, $\sigma_0$ is the surface normalised backscattering cross section, $\Gamma$ its Fresnel reflection coefficient and $A_{surface\rightarrow target}$ is the one-way attenuation encountered in the path between

the surface and the target expressed in dB. Therefore the mirror return is reduced by two factors:

1. the attenuation due to the extra path (four times the path from the target to the surface) expressed by the second term on the right hand side of Eq. (1);

2. an additional "mirror loss factor", the third term on the right hand side of Eq. (1), which accounts for the specular reflection property of the surface, its backscattering cross section and the geometry of observation.

When considering the parameters of the EarthCARE radar the mirror loss factor assume the values plotted in Fig. 1 for characteristic values of $\sigma_0$ between 5 and 18 dB and target altitudes from 0 up to 14 km. Very close to the surface the mirror loss is at a minimum level and converges to the value $10\log_{10}(\Gamma^4) = -8.5$ and is independent on $\sigma_0$ and the observation geometry. Generally for water surfaces, this limit value ranges between -7.6 and -10 dB since $\Gamma$ assumes values between 0.56 and 0.64 with temperature between 0 and $20°C$, respectively. Note that this (minimum) mirror loss is larger in magnitude than

at smaller frequencies, e.g. -4.4 dB at Ku and Ka (Li and Nakamura, 2002). When increasing the height of the target the mirror losses tend to increase and increasingly show a dependence on $\sigma_0$. This tendency depends on the relative weight of the two terms in the denominator of the third term on the right hand side of Eq. (1): if the first term dominates the mirror loss becomes equal to $\Gamma^4$; otherwise it becomes proportional to $\sigma_0$, to $\Gamma^2$, to $\theta_{3dB}^2$ and to the square of the ratio $H_{sat}/H_t$. The green lines in Fig. 1 highlight where the two terms are equal for the two configurations considered in this paper. Because of the different

satellite heights and radar beamwidths the mirror losses for the EarthCARE radar tend to be higher than for the CloudSat radar (compare black dashed and continuous contour lines), with the weight of the first term being generally larger in CloudSat configuration when the same $\sigma_0$ and $H_t$ are considered (compare green lines).

When passing to reflectivities expressed in dBZ, Eq. (1) becomes:

$$Z(r_m) = Z(r_t) + 20\log_{10}\frac{r_m}{r_t} - 4\,A_{surface\rightarrow target} + 10\log_{10}\left[\frac{(H_{sat}-H_t)^2\Gamma^4\sigma_0}{\sigma_0 H_{sat}^2 + 11.04\Gamma^2\frac{H_t^2}{\theta_{3dB}^2}}\right] \tag{2}$$

with a slight amplification term on the right hand size due to the difference in ranges between $r_t$ and $r_m$. Though the mirror losses can be quite substantial (up to -30 dB, but more typical values are in the range between -10 and -25 dB) it is clear that, because of the enhanced sensitivities of W-band radars, they are not sufficient to push the mirror signals below the noise level (-35 dBZ and -29 dBZ for EarthCare and CloudSat, respectively). For instance, when considering a strong reflecting ice clouds (15-20 dBZ) located at around 10 km in the EarthCARE configuration a total reduction of more than 50-55 dB is needed to

be sure that such cloud will not produce second trip echoes. Since gas attenuation never exceed 8-9 dB two ways even in very



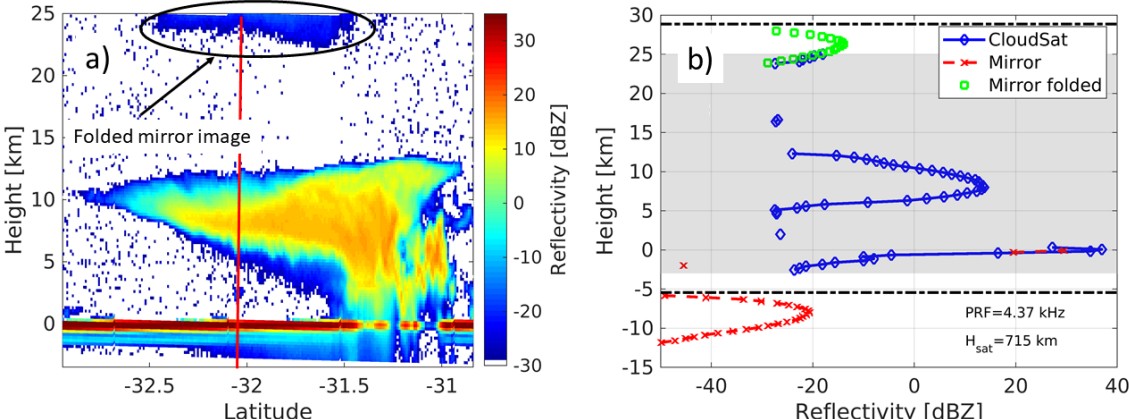

**Figure 2.** Panel a: example of mirror image in CloudSat data produced by a high reflectivity tropical anvil cloud. The mirror image folds back at heights between 22 and 25 km. Panel b: exemplification of the folding mechanism for the profile at -32° latitude marked with the red line in the left panel. The black dash-dotted lines delimit the folding ranges; the grey shaded region covers the CloudSat data window.

humid and warm atmospheres, this requires additional contributions from hydrometeor attenuation. If such extra attenuation is not present (no low/mid level clouds), then a mirror image is likely to cause second trip echoes in typical conditions of operation of space-borne W-band radars.

## 2.1 Example of mirror images in CloudSat data

This situation is epitomized by the example of a mirror image producing second trip echo for the CloudSat observations of an anvil clouds with a highly reflective core located at about 10 km (Fig. 2a). The folding mechanism is further explained in Panel b for the reflectivity profile at -32° latitude (corresponding to the red line in Panel a) in correspondence to which the satellite height is 715 km and the PRF is 4.37 kHz (as recorded in the Level-1 CPR product). By distributing an integer number of 34.3 km long unambiguous ranges from the satellite height downwards, the folding window (delimited by the dash-dotted

black lines) for this profile is between 29 and -5.3 km (but note that CloudSat records data only below 25 km and above -3 km, grey shading). The mirror image corresponding to the reflectivity profile is computed according to formula (2) with the value of $\sigma_0$ derived according to Tanelli et al. (2008) and the value of the attenuation derived from the gas attenuation profile. The mirror image produce reflectivities roughly 20 dB lower than the anvil for ranges between -5 and -12 km below the surface (red line). When this signal is folded back into the folding window it produces a signal well above CloudSat sensitivity from

29 down to 22 km (green squares). This is partially recorded by the radar sampling window, with the measured reflectivities reaching values comparable with the simulated ones (compare green and blue curves). This example gives confidence that the simulation of the mirror and its folding is producing realistic results.





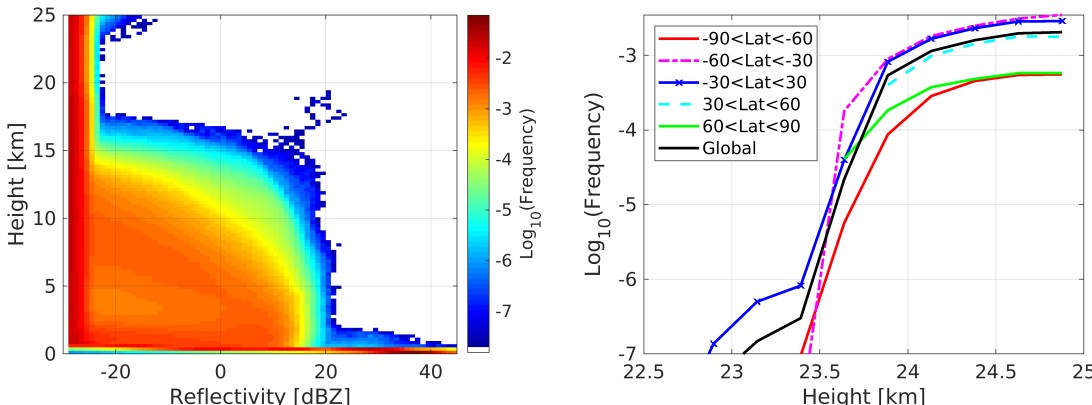

**Figure 3.** Left panel: CFAD of CloudSat reflectivities over the ice free ocean with no convection for 2008. The anomalous presence of clouds between 23 and 25 km with reflectivities between -28 and -20 dBZ is due to second trip echoes associated to mirror images. Right panel: frequency of second trip echoes above 22 km for different latitudinal bands derived from the CFAD substracting the "noise floor" as computed by averaging altitudes between 19.5 and 20.75 km.

## 2.2 Climatology of second trip echoes associated to mirror images in CloudSat database

The presence of second trip echoes in CloudSat is confirmed statistically by looking at the contour frequency altitude display (CFAD) of CloudSat CPR data for the full 2008 (left panel in Fig. 3). Here we have only considered profiles over open ocean with no sea ice (as determined by the CloudSat 2C-PRECIP-COLUMN, Haynes et al. (2009)) and not in presence of convection (as derived from the CloudSat 2B-CLDCLASS product, Sassen and Wang (2008)); this roughly correspond to 51.5% of the CloudSat profiles, 15.2%, 91%, 68%, 48%, and 15.3% of the CloudSat profiles in the different zonal bands as indicated in the legend. The presence of fake clouds between 23 and 25 km with reflectivities between -28 and -20 dBZ confirms the presence of second trip echoes associated to mirror images. Globally, when considering the layer between 19.5 and 21 km as a reference for returns with no second trip echoes, this corresponds to fake clouds with a frequency of less than one out of 500 CloudSat profiles at 25 km (right panel in Fig. 3). Note that the occurrences very quickly drop to less than one profile out of one million below 23.5 km. Higher frequency are registered in the tropical belt (blue line) and in the Southern mid-latitudes. Because of their location at height well above the tropopause, it is very easy to flag these clouds and exclude them from further analysis.

## 2.3 EarthCARE folding

EC will adopt substantially higher PRFs than CloudSat, with interlaced low and high PRF modes in order to optimize the Doppler performances (Tomiyama et al., 2020). Because of the change of altitude of the satellite along the orbit, round trip time will also change along the orbit and these changes may exceed the PRI. This requires to design the radar timing via

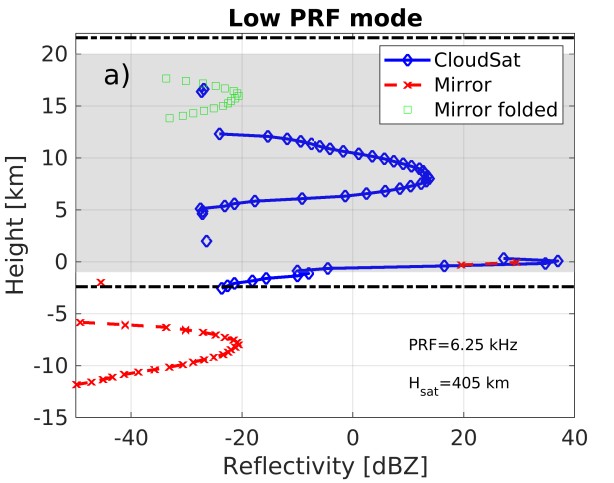
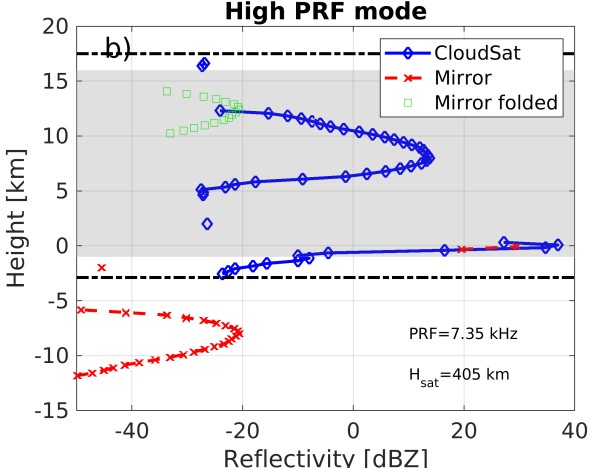

**Figure 4.** Example of how the mirror image is expected to fold for the low (Panel a) and high (Panel b) PRF mode for the EarthCARE CPR in correspondence to the same CloudSat profile illustrated in Fig. 2b. The black dash-dotted lines delimit the folding ranges; the grey shaded region covers the EarthCARE sampling window for the two modes.

variable PRF so that the radar echo window remains centered within the troposphere. Here to study the effect we only analyze two situations with the satellite always flying at the same orbit (405 km):

1. a "low PRF" mode with a PRF of 6.255 kHz ($r_u = 24$ km), a folding interval from 21.6 and -2.4 km and a sampling window between 20 and -1 km;

2. a "high PRF" mode with a PRF of 7.35 kHz ($r_u = 20.4$ km), a folding interval from 17.5 and -2.9 km folding and a sampling window between 16 and -1 km.

The same CloudSat profile illustrated in Fig. 2 will fold as depicted in Fig. 4 for these two modes. We can clearly distinguish two situations.

1. The mirror image is folded back into a region with no cloud (as detected by CloudSat). This will originate additional

fake cloud cover; this is the case for the "low PRF mode", Panel a.

2. The mirror image is folded back into a region where there is already a cloud; this is the case for the "high PRF mode" (Panel b) where part of the second trip echo overlaps with the first trip echo. In this case the mirror image will both slightly extend the cloud cover above the real cloud top and will modify the value of the real cloud reflectivity but only if the signal-to-mirror ratio (SMR) is small (e.g. if SMR=0 dB like at 11.3 km it will double the signal).

15 In both cases the situation looks much more ambiguous than the second trip echo generated by CloudSat and it will require much more effort to be properly identified.

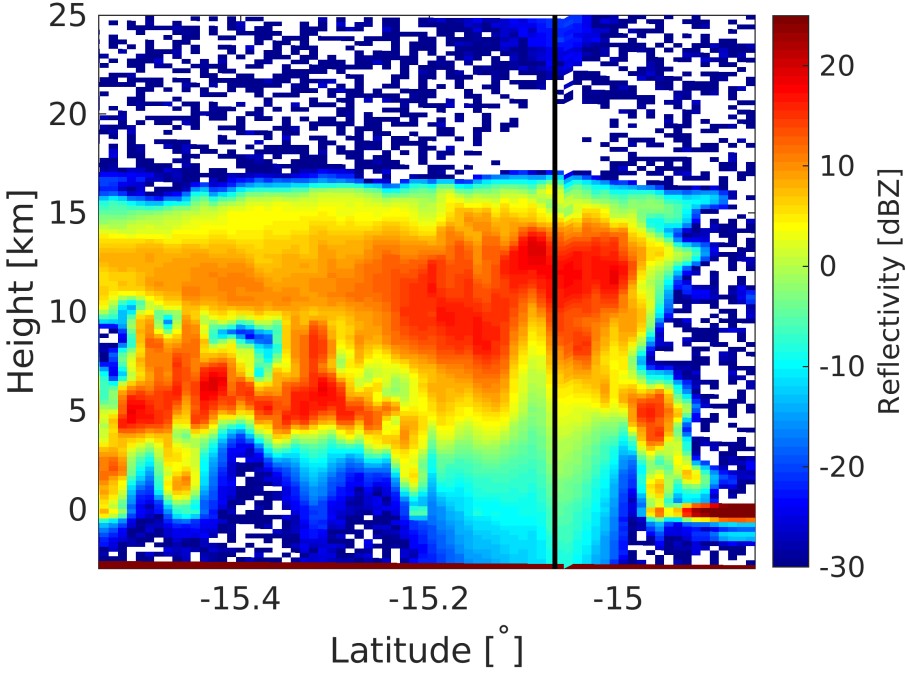

**Figure 5.** Example of second trip echoes, visible between 22 and 25 km, generated by MS tails in CloudSat reflectivity vertical profile observations of a deep convective system. The satellite is moving southward from Indonesia towards Australia over the Indian Ocean.

## 3   Multiple scattering tails and their simulation

In presence of deep convection MS can be so strong that it can generate returns from ranges well below the surface by producing long tails, a phenomenon known as pulse stretching (Hogan and Battaglia, 2008). The typical profiles in such condition tend to peak close to 20 dBZ at very high altitudes ($> 10$ km) and then have reflectivities decreasing towards the ground. Very often the surface return peaks corresponding to the surface tend to disappear. A case study is shown in Fig. 5 for a deep convective system occurred on the 11th January 2008 over the Indian Ocean North of Western Australia. The system reaches above 17 km in height; the convective core, where very strong MS clearly occurs, extends for almost 25 km along track from -15.2° to -14.95°. In that region, when focusing on surface ranges, the surface peak always disappears while the reflectivity signal goes below the minimum detectable level only on the right flank and it maintains values as high as -6 dBZ in the center of the core.

The sloping of all the reflectivity profiles in the core presents very low values and tends to become smaller getting closer to the surface where they can reach values lower than 1 dB/km. This is clearly incompatible with single scattering which predicts such low values already in correspondence of rainfall rates of 0.6 mm/h (Matrosov et al., 2008).

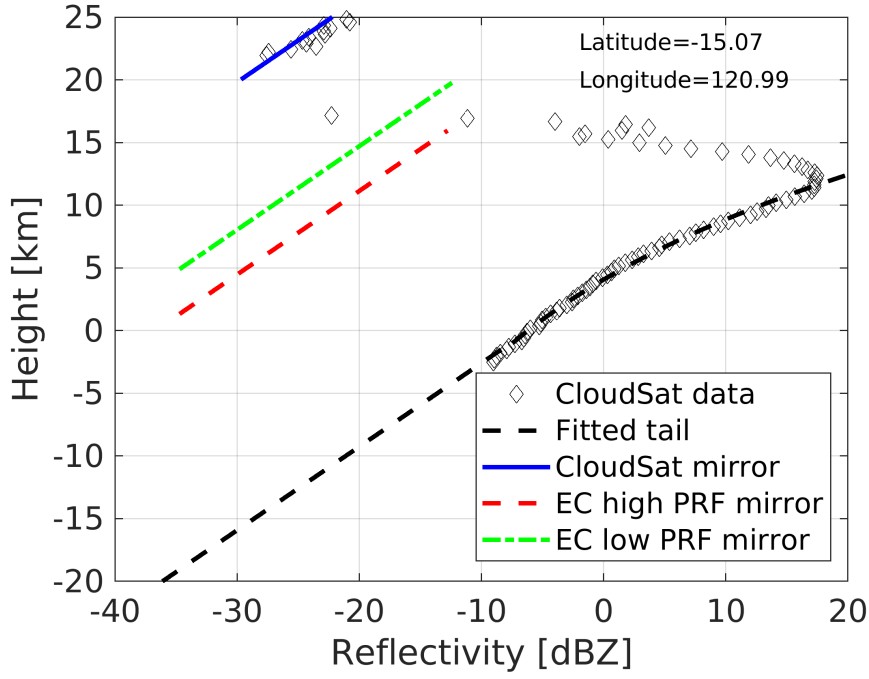

**Figure 6.** Example of folding of MS tails for the profile corresponding to the black line shown in Fig. 5. The CloudSat reflectivity profile (diamonds) which is sampled only down to -2.5 km is extrapolated down to -20 km (dashed black line) and then folded back according to the CloudSat unambiguous range (blue line) and the two EarthCARE high and low PRF modes (red and green line, respectively).

In a very crude approximation we have extrapolated the MS tail by using an interpolation of the decreasing part of the measured reflectivity profile (excluding the surface peak, if present) by an exponential function of the form:

$$Z_{fit}(z) = A + B \exp(C\,z) \tag{3}$$

with $A$, $B$ and $C$ as fitting parameters; however, we have altered this function to a line with slope of 1.5 dB/km once the slope
5 of the exponential function surpasses this limit. An example of the fitting procedure is demonstrated in Fig. 6 where we have used the reflectivity profile corresponding to the black line shown in Fig. 5, acquired by the CloudSat radar circa 250 km from the Dampier peninsula in Western Australia. The CloudSat reflectivity measurement, plotted with diamonds, clearly indicates the presence of second trip echoes with reflectivities values between -27.6 and -21 dBZ between 22 and 25 km. These are well explained if the extrapolated MS tail (dashed black line) is folded back according to the CloudSat satellite height and PRF (blue
10 line). The effect is the generation of a fake cloud, the reflectivity of which remains marginally above the CloudSat minimum sensitivity threshold.

If we fold back the same MS tail by using the EarthCARE high and low PRF modes described in Sect. 2.3 the folded signals (red and green line, respectively) remain above the EarthCARE minimum sensitivity for a much longer distance than for the





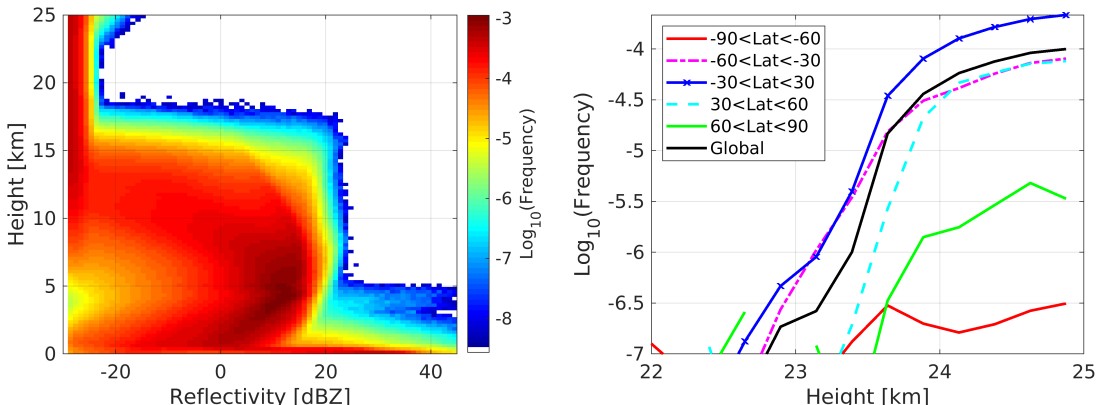

**Figure 7.** Left panel: CFAD for CloudSat profiles in convection for 2008 and 2009. The appearance of second trip echoes is clearly visible above 23 km. Right panel: frequency of second trip echoes for convective profiles as a function of height for different latitudinal bands as indicated in the legend.

CloudSat case; however, with diverse outcomes: whereas with the low PRF mode a fake cloud will appear between 17 and 20 km (the sampling window upper limit) with the high PRF mode the second trip echoes will be completely overwhelmed by the first trip echo and will only marginally modify the sampled signal at 16 km since the second trip echo is 15 dB lower than the first echo signal. This outcome of course depends on the interplay between height of the system, adopted PRF mode and
sampling window.

## 3.1   Climatology of second trip echoes associated to MS tails in CloudSat database

Convective profiles have been extracted by using the classification of the 2B-CLDCLASS product for the full 2008 CloudSat dataset. Circa 1.1% of all profiles are classified as convective. The corresponding CFAD is depicted in the left panel of Fig. 7. As in Sect. 2.2 second trip-echoes appear at the very top of the CloudSat sampling window above 23 km. The frequency of
such echoes as a function of height for different latitudinal bands is shown in the right panel. The frequency is maximised at 25 km and strongly decreases moving towards the poles. The phenomenon is rare; in fact even at its maximum (at 25 km and for the Tropical belt) basically it occurs with an absolute frequency of $10^{-3.7}$, i.e. roughly one out 50 convective profiles presents a second trip echo.

## 4   Second trip echoes in EarthCARE observations

The CloudSat database can be used to predict the occurrence of second trip echoes associated to mirror mages and to multiple scattering tails.

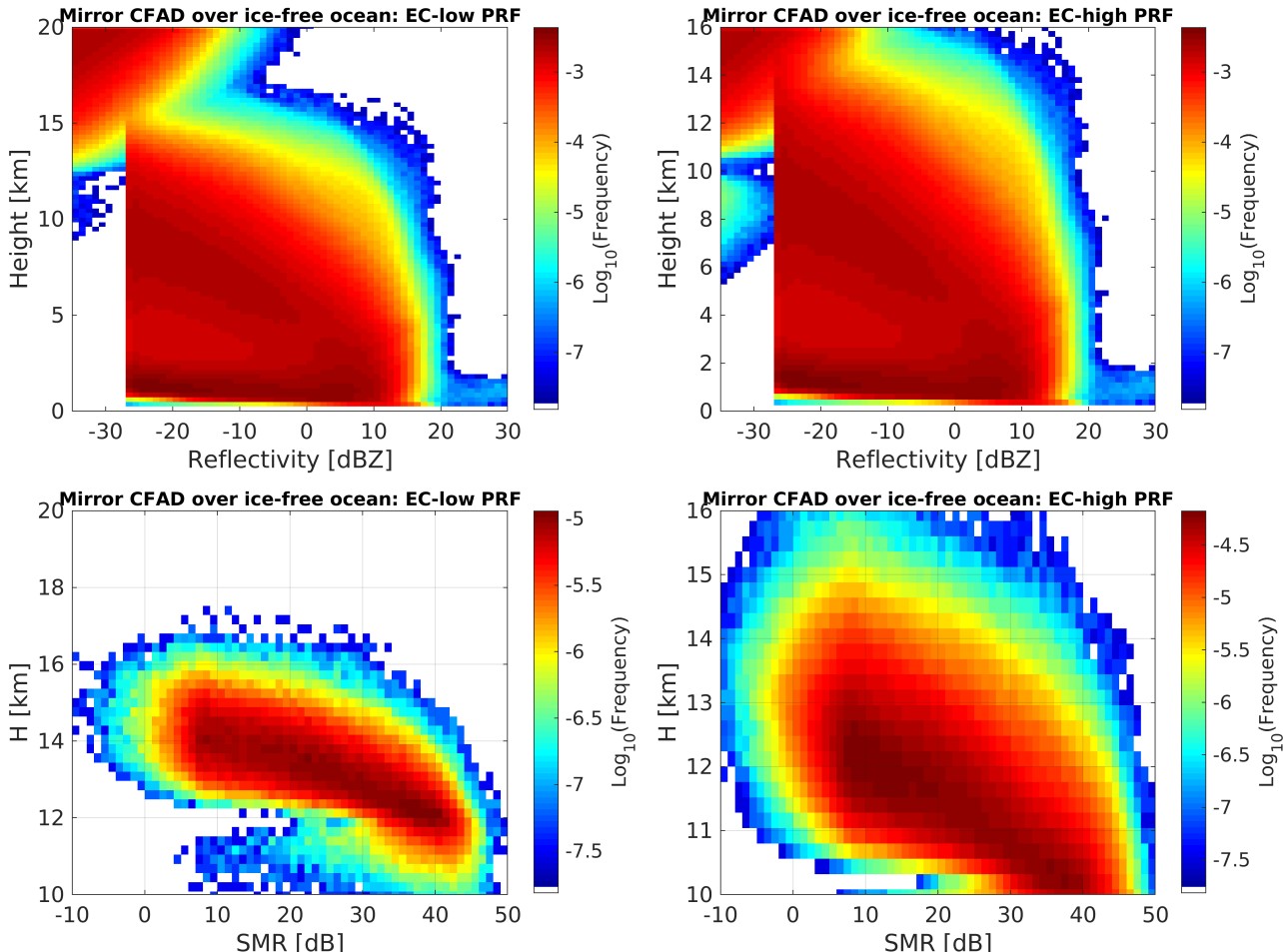

**Figure 8.** Expected impact of second trip echoes generated by mirror images in EC observations. Top panels: simulated CFAD of reflectivities for the "low PRF" (left) and "high PRF" (right) EarthCARE configurations. Bottom panels: same as top panels for simulated CFAD of signal to mirror ratios (SMR).

### 4.1    Second trip echoes associated to mirror images

The CloudSat profiles are used first to simulate mirror images by using Eq. (2) and the EC parameters for satellite height (405 km) and beamwidh (0.095°); then the mirror images are folded according to the the two EC configurations as explained in Sect. 2.3. Only profiles over the ice-free ocean and with no convection are considered. The CFADs for the "low" and "high PRF" modes are shown in Fig. 8 (top panels). Differently from CloudSat, the presence of second trip echoes is now contaminating the region well below the tropopause for both modes. A primary peak appear in correspondence to the top of the sampling window extending for several kilometers whereas a secondary peak is present with less pronounced reflectivities at





much lower altitudes centred around 11 and 9 km for the low and high PRF modes, respectively. Since the reflectivity CFADs represent a cumulative frequency it is not possible to understand whether the second trip echoes occur in correspondence to an already existing cloud or do represent a fake cloud. We have therefore produced separated CFADs for second trip echoes which do not correspond to a signal strong enough to be identified as a cloud by the 2B-CLDCLASS algorithm and which occur

concomitantly with a real cloud (not shown). In the latter situation we have also computed SMRs, i.e. the signal to mirror ratios between the cloud true signal and the second trip echo fake signal; their CFADs are depicted in the bottom panel of Fig. 8. Clearly the majority of the second trip echoes have strength much lower than the real cloud signals but a not negligible portion of second trip echoes do produce signals which overwhelm the direct cloud signal. This is the case only at heights between 12.5 and 16.5 km (10.5 and 16 km) for the low-PRF (high-PRF) mode.

In Fig. 9 the frequency of second trip echoes alone (top row) and for coincident second and first trip echoes (bottom row) for the "low PRF" (left) and "high PRF" (right) EC configurations are shown as a function of the height. With the high PRF mode the second trip echoes stretch to lower altitudes (compare top panels). For both modes, at the top of the sampling window, the expected absolute frequency of such instances is about 3% globally (or roughly 6% over the ice free ocean). This frequency is rapidly decreasing to less than 0.1% at 14.5 km (11.8 km) for the low (high) PRF mode.

The frequency of coincident first and second trip echoes (Fig. 9) is generally much lower peaking at around 13.5 and 11.5 km for the low and high PRF mode, respectively. This second trip echoes will enhance the cloud reflectivity signal but in a perceptible way only when the SMR is lower than 3 dB. This happens with frequencies less than $10^{-5.2}$ and $10^{-4.1}$ even in correspondence to the height with the highest number of occurrences (14.2 and 12.8 km) for the low and high PRF mode, respectively.

**4.2 Second trip echoes associated to multiple scattering tails**

A similar procedure has been adopted to simulate second trip echoes associated to MS tails. Out of profiles classified as convective by the 2B-CLDCLASS product only those with a surface return lower than 0 dBZ and reflectivity in the ice layer exceeding 14 dBZ are used as candidates to produce MS tails. Only 1.1% of the profile are classified as convective and only 0.07% fulfill the previous MS tail condition. These profiles are also strongly concentrated in the Tropics (86%), with much

fewer (5 and 9%) in the Southern and Northern mid-latitudes and extremely scant percentages in the Northern high-latitudes (0.2%). No profiles are found in the Southern high latitudes.

The fitting procedure described in Sect. 3 is used to extrapolate a tail which is then folded according to the two EC modes (Sect. 2.3). No MS tail is produced otherwise. As before, CFADs of reflectivities and SMR for the two modes are produced (Fig. 10). Differently from CloudSat (compare with Fig. 7) and similarly to what seen with the mirror images (Fig. 8) now there

is a considerable overlap between first and second trip echoes. In this case, as the bottom panels of Fig. 10 show, any significant overlap between first and second trip echoes is practically absent in the low PRF mode and marginally present only between 13 and 16 km in the high PRF mode, where very rarely the SMRs are smaller than 3 dB. This is confirmed by Fig. 11: the most prominent effect associated to MS tails is indeed the appearance of fake clouds with frequencies exceeding one out of 3000 profiles at the top of the sampling window for both EC-modes. The polar regions (and particularly the Southern hemisphere)

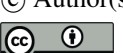

**Figure 9.** Simulated frequency of second trip echoes alone (top row) and for coincident second and first trip echoes (bottom row) both above the minimum detection threshold for the "low PRF" (left) and "high PRF" (right) EarthCARE configurations. In the bottom panels three curves are presented corresponding to signal-to-mirror ratio (SMR) lower than 0 and 3 dB (dashed blue and continuous black lines, respectively) or for any SMR (dash-dotted red line).

have a scarcity of such events; on the other hand, in the Tropics (blue line) the occurrence of such events exceed 1 out of 1000 cases. For the heights with the highest occurrences only one profile out of circa 50,000 (10,000) has the potential to produce a the second trip echo due to MS tail that will modify the reflectivity of a real cloud return.







**Figure 10.** Same as Fig. 8 for the second trip echoes generated by MS tails.

## 5   Conclusions and future work

Second trip echoes generated by mirror images over the ocean and multiple scattering tails in correspondence of deep convective cores can become an issue in W-band cloud radar space-borne observations. In CloudSat observations they do represent a rarity that tend to appear above 20 km; as a result such features can be easily screened out and indeed they have passed almost undetected by the cloud radar community. However things may change for other W-band space-borne system which are envisaged to adopt PRFs much higher than the one used by the CloudSat CPR (from 3.7 to 4.4 kHz) in order to improve their Doppler capabilities.

In this work CloudSat observations and Level 2 products have been used to simulate the impact of second trip echoes in the up-coming EarthCARE observations. We have used two configurations with a low-PRF (6.3 kHz) and a high-PRF mode

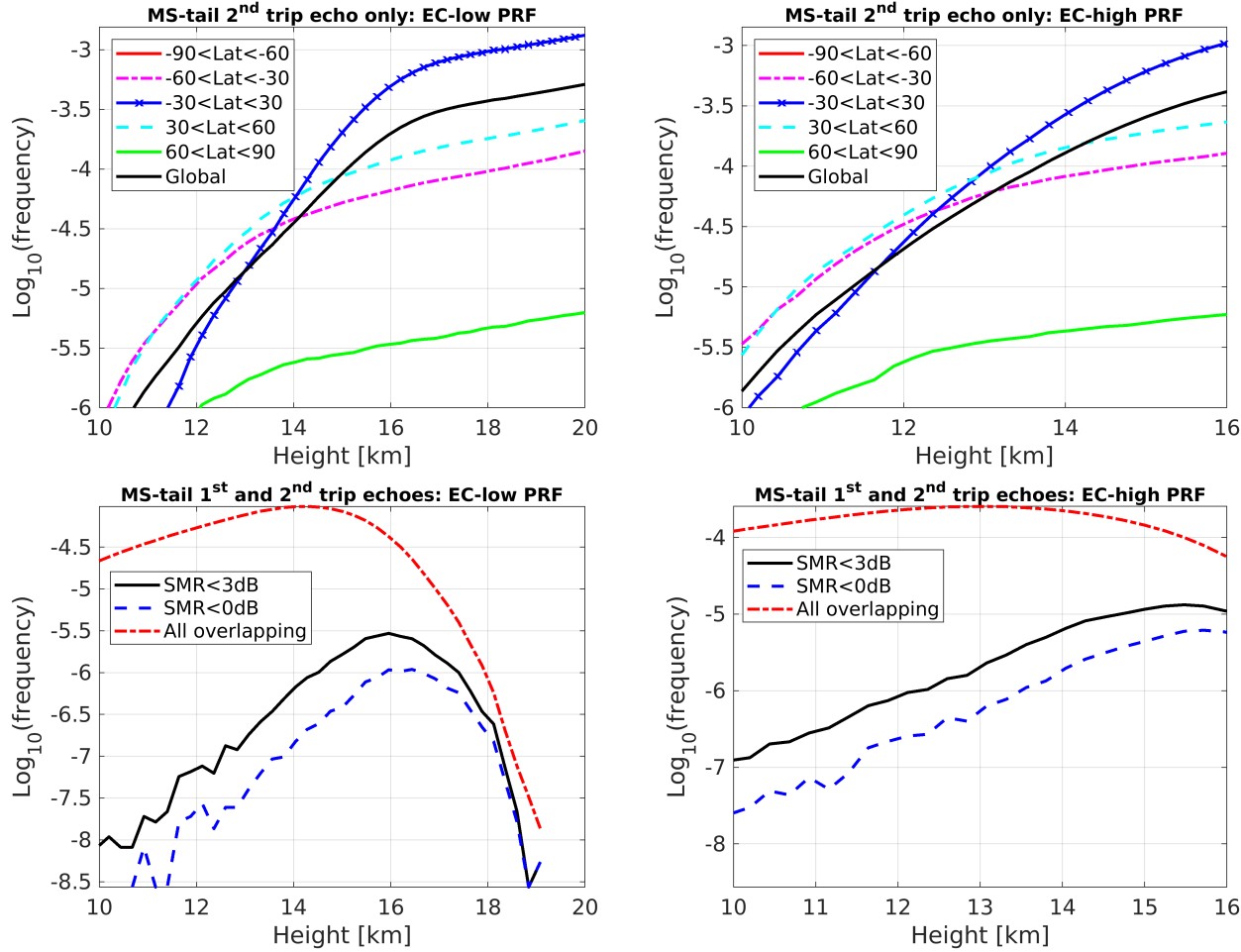

**Figure 11.** Same as Fig. 9 for the second trip echoes generated by MS tails.

(7.3 kHz) as an example of the general behaviour expected for the EC CPR. Our findings show that the presence of such echoes cannot be neglected: in particular, over the ocean, mirror images will tend to populate the EarthCARE sampling window with a maximum frequency at the top of the sampling window. This will create additional fake cloud cover in the upper troposphere (of the order of 3% at the top of the sampling window and steadily decreasing moving downwards) and, in much less frequent

5    instances, it will cause an amplification of signals in areas where clouds are already present. MS tails will produce also second trip echoes but with much lower frequencies: less than one profile out of 1000 in the Tropics and practically no effects at high latitudes.

    At the moment Level-2 algorithms of the EarthCARE CPR do not account for such occurrences. They will have to properly remove these second trip echoes and to correct for reflectivity enhancements, where needed. This task can be facilitated by the





fact that the amplitude and location of second trip echoes can be predicted. In addition to this, the EC High Spectral Resolution Lidar co-located measurements will be able to unambiguously identify fake echoes as well.

This work is relevant for the design of future space-borne Doppler radar missions as well. The use of high PRF mode aimed at improving Doppler performances (Kollias et al., 2014; Sy et al., 2014; Hagihara et al., 2021) must be considered

cautiously. Multiple PRFs could be adopted to separate first and second trip echoes like routinely done for ground-based precipitation radars; however, for low orbiting satellites, constraints related to the change of altitude of the satellite along the orbit (Tomiyama et al., 2020) or to the specific design of the Doppler measurements (e.g. Durden et al. (2007)) may significantly reduce the possible combinations of multiple PRFs. The most straightforward way to solve the Doppler dilemma in space-borne system remains in our view the use of polarization diversity (Kobayashi et al., 2002; Battaglia et al., 2013; Wolde et al., 2019).

The experience with CloudSat clearly demonstrate that, for nadir looking radars, PRFs of the order of 4 kHz are good enough to reduce the presence of second trip echoes to mere isolated cases, easy to be spotted, flagged and eliminated when building cloud statistics. For slant looking radars like proposed by the Wivern mission (Illingworth et al., 2018) the mirror effect can be completely neglected because of the slant viewing geometry; MS tails could produce second trip echoes but likely less than here predicted for EC again because of the viewing geometry.

The current modelling is based on the simple approximation of the mirror image proposed by Meneghini and Atlas (1986). More sophisticated modelling e.g. based on MonteCarlo methods (Battaglia and Simmer, 2008) could account for contributions from higher order of scattering as well (which will exacerbate the impact). Specific orbital and variable PRF modelling for the EarthCARE analysis could also be considered, though the overall message is not expected to change. Similarly a better description of the MS tail could be provided by 1D radiative transfer code (e.g. Hogan and Battaglia (2008)) though it is clear

that three-dimensional effects play a role in affecting the strength of the MS tail.

A final consideration: since the accurate modelling of the second trip echoes and their amplitudes require a proper quantification of the scattering properties of the layers generating the MS tails and of the attenuation between the mirror target and the ground, the measurements of the second trip echoes could represent an opportunity, e.g. to be exploited for providing additional constraints about the hydrometeor properties.

*Acknowledgements.* This research used the ALICE High Performance Computing Facility at the University of Leicester.

*Data availability.* All CloudSat data are freely available at http://www.cloudsat.cira.colostate.edu/data-products

*Author contributions.* AB is the only author.



*Competing interests.* The authors declare that they have no conflict of interest.





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
