# Peer review of "Impact of second trip echoes for space-borne high PRF nadir-looking W-band cloud radars"

_Atmospheric Measurement Techniques, 2021_

## Author Response (AR1)

**Reply to reviewer 1**

We thank the reviewer for the useful comments.

**Page 16, Line 21: could you elaborate on how you think the mirror echoes add any information. I think the fact that you can predict the magnitude and location of the mirror echoes with some accuracy suggests that there isn't much additional information**

In this paper we have simulated mirror image to the ``best of our knowledge''. A correct simulation of mirror images requires a correct estimate of sigma_0 and of the attenuation between the mirror and the ground. As a result, checking that measured mirror images reflectivities are properly simulated is an indirect check on sigma_0 and PIA estimates. In principle in (light) rain mirror echoes could be used to cross check whether the attenuation correction algorithm is producing a realistic attenuation profile . A sentence along these lines has been added at the end of the paper (page 16 lines 33-35).

**Specific Comments:**

**Page 1, line 22: phase 0 -> pre-phase A**

DONE

**Page 5, Line 6: The two sentences beginning with 'The folding' are not clearly written. I Suggest: *'The folding mechanism is further explained in Panel b for the reflectivity profile at -32 latitude (corresponding to the red line in Panel a). The Level-1 CPR product reports the satellite height is 715 km and the PRF is 4.37 kHz which corresponds to an unambiguous range of 34.3 km. By distributing an integer number of these 34.3 km long unambiguous ranges from the satellite height downwards, the folding window'***

*DONE*

**Page 6, Line 5: It isn't labeled in the legend which percentage goes with each latitude band.**

NOW SPECIFIED (Lines 11-13 page 6)

**Page, Line 11: It's also pretty straightforward to use CALIPSO to identify these clouds.**

ADDED Page 7 lines1-2

**Reply to reviewer 2**

We thank the reviewer for the useful comments.

**Could the Autor explain why only CloudSat profiles over the ocean considered for the statistical analysis?**
The statistical analysis is restricted to ocean surfaces only for the mirror echoes and this is because mirror images are strongly suppressed over land surfaces, which are rougher and therefore do not behave as Fresnel surfaces. Therefore we do not expect second trip echoes generated by mirror images over land.

A new sentence has been added lines 6-8 page 6

**And is it also possible to do such detection and analysis for profiles overland?**

The statistical analysis for the multiple scattering tails on the other hand has not been restricted to the ocean only but it has been applied to any surface (and indeed convection is generally stronger over land than over ocean). In presence of strong multiple scattering the surface is not playing any role because the radar transmitted signal is actually not reaching it! See new sentence at Line 26-28 Page 9.

**Reply to reviewer 3**

We thank the reviewer for the useful comments.

**As someone who would like to repeat these simulations so that I understand the math better, can the constants used to generate Figs. 1 and 2b be added to the figure caption or in the text? If I can repeat Fig. 2b, then I could repeat the EarthCare simulations.**

The constants have now been introduced in the caption

**Also, I may not understand how profiles with MS scattering are being counted (page 10 line 12). Should a frequency of 10^(-3.7) be one on 5000 profiles (not one in 50 profiles)? Maybe 10^(-3.7) is relative to all profiles in 2008.**

10^-3.7 is an absolute frequency to all CloudSat profiles. So 1 profile out of 5000 profiles present this MS tail in the tropical belt. Since convective profiles are roughly 1% of the profiles this correspond roughly to one profile out of 50 convective profiles. So there is a 2% of the convective profiles which produce some second trip echoes. We have rephrased the sentence to make it more clear (line1 page10).